# The Impact of Long-Term Care Insurance for Older Adults: Evidence of Crowding-In Effects

**DOI:** 10.3390/healthcare13121357

**Published:** 2025-06-06

**Authors:** Hyeri Shin

**Affiliations:** Department of Gerontology, AgeTech-Service Convergence Major, Kyung Hee University, Yongin-si 17104, Gyeonggi-do, Republic of Korea; zisoa@khu.ac.kr

**Keywords:** long-term care insurance, crowding-in effects, crowding-out effects, older adults

## Abstract

**Objectives:** This study investigates the presence of crowding-in or crowding-out effects of long-term care insurance (LTCI) on older adults’ care in Korea. Additionally, it examines the influence of old-age income security and private systems, including private transfer income and private health insurance, on these effects. The analysis focuses on three aspects: family-provided care, private non-family care, and total care expenses. **Methods:** This study conducted logistic and linear regression. Logistic regression was used to examine the likelihood of receiving family-provided and private non-family care, while linear regression analyzed factors associated with total care expenditures. **Results:** The results reveal a crowding-in effect for family care, as greater utilization of public LTCI is positively associated with family-provided care. However, the relationship between public LTCI and private non-family care was not statistically significant, suggesting that the crowding-in effect on private care systems remains limited. Lastly, LTCI utilization was significantly associated with higher care expenditures. It is noteworthy that the current public LTCI in Korea has low coverage, resulting in insufficient care provision. Consequently, there is growing activity in the private care sector. **Conclusions:** These findings highlight the need for a more integrated approach to long-term care in Korea, balancing public, private, and family care resources. To achieve quality integrated long-term care for older people, policymakers should focus on expanding public LTCI coverage while fostering coordination between family caregivers and professional care services, ensuring a comprehensive and high-quality care system that meets the diverse needs of Korea’s aging population.

## 1. Introduction

The accelerating global demographic shift toward population aging has heightened the urgency of developing sustainable and equitable long-term care (LTC) systems. As the prevalence of disability and chronic health conditions increases, traditional family-based caregiving models are becoming increasingly strained, and the demand for both formal and informal care is expected to surge [1,2,3]. In response, many countries—including Korea, Japan, and China—have introduced or expanded long-term care insurance (LTCI) programs aimed at alleviating family caregiving burdens and reducing unnecessary institutionalization [1,2,4,5]. In Korea, the public LTCI system was introduced in July 2008 to address the rapidly growing needs of its aging population. The number of individuals certified for LTCI benefits grew from 770,000 in 2019 to 860,000 in 2020, 950,000 in 2021, and exceeded 1,010,000 in 2022. In 2023, the number of recipients who actually used LTCI services reached 1,073,452, marking a 7.4% increase from the previous year [6]. The coexistence of traditional informal care and the expanding role of formal LTC services has led to complex interactions, particularly regarding the “crowding-in” and “crowding-out” effects, where public LTCI may either complement or substitute family-provided and privately provided care [7,8,9,10].

Understanding the mechanisms of crowding-in and crowding-out is central to analyzing the evolving landscape of LTC systems. “Crowding-out” refers to situations where the introduction or generosity of public LTCI leads to a reduction in informal caregiving efforts, typically as formal care substitutes for previously unpaid familial support [8,10]. For example, families may reduce the intensity of caregiving, particularly for physically demanding tasks such as bathing, transferring, and toileting—after their family member begins using LTCI services. In contrast, “crowding-in” occurs when formal LTC programs enhance informal caregiving by alleviating caregiver burdens, expanding the pool of caregivers, or improving the sustainability of family support networks [7,11]. For example, the availability of formal LTCI services may enable families to continue providing emotional support and household assistance while relying on professional caregivers for more intensive tasks, thereby sustaining overall caregiving involvement. Importantly, these effects are not mutually exclusive and may co-occur depending on policy design, cultural norms, and institutional contexts. For example, studies from Europe show that generous formal LTC systems are associated with a higher prevalence of individuals providing some level of informal care (crowding-in) but a reduced intensity of caregiving (crowding-out of the most burdensome tasks) [7]. Thus, the extent and nature of crowding effects reflect the specific configuration of LTC benefits, societal expectations regarding family obligations, and the broader caregiving environment [7,8,12].

Empirical findings from different national contexts further highlight the nuanced impact of LTCI on caregiving dynamics. Research in European countries found evidence of family crowding-out effects, although no robust evidence of public sector crowding-out was observed [13]. In contrast, studies on China’s pilot LTCI program reported a clear crowding-in effect, with a 17.2% increase in family caregiving following program implementation [9]. In the South Korean context, research has demonstrated that the introduction of LTCI improved health outcomes among disabled older adults and significantly reduced healthcare expenditures for beneficiaries compared to non-beneficiaries [5]. However, the effects on the intensity of family caregiving and the labor market participation of family members remain areas of active debate and ongoing investigation. Addressing these gaps, the present study systematically examines the determinants and interplay of family caregiving, paid caregiving, and care-related expenditures under a multi-tiered security system, with particular attention to how LTCI integration shapes caregiving responsibilities and cost dynamics within Korea’s unique cultural and demographic context.

With the introduction of public long-term care insurance (LTCI) in Korea in 2008, the care system for older adults has undergone a significant shift. Empirical studies indicate that while the expansion of formal LTCI services has substantially increased the use of institutional and community-based care, a partial crowding-out effect on informal care has also been observed [5]. Specifically, formal services have substituted for certain aspects or intensity levels of family-provided caregiving, particularly at the intensive margin [14]. Nevertheless, the overall likelihood that care would be provided—whether by family members or formal providers—remained largely unchanged [3,14].

The present study aimed to empirically examine how the utilization of public long-term care insurance (LTCI) influences caregiving patterns and financial burdens among older adults in Korea. Using nationally representative data from the Korean Longitudinal Study of Ageing (KLoSA), I analyzed the relationships between LTCI use and three key outcomes: family-provided care, private non-family care, and the level of total care expenditures. This investigation contributes to a deeper understanding of the crowding-in and crowding-out effects of formal care services within Korea’s long-term care system.

## 2. Materials and Methods

### 2.1. Study Population

This study utilized data from the 9th wave of the Korean Longitudinal Study of Ageing (KLoSA), conducted by the Korean Employment Information Service (https://survey.keis.or.kr, accessed on 29 April 2025). The dataset is a nationally representative sample approved by Statistics Korea (Approval No. 336002). Of the total 6057 respondents, individuals under the age of 65 (*n* = 1566) and those with missing values in key variables for this analysis (*n* = 1388) were excluded, resulting in a final sample of 3103 individuals aged 65 and older.

### 2.2. Dependent Variables

To examine care patterns and financial burden among older adults, three models were constructed, each with a different dependent variable. In Model 1, the dependent variable was family-provided care, coded as 1 if the primary caregiver was a family member and 0 for all other caregiving types. In Model 2, the dependent variable was use of private non-family care, coded as 1 if the individual utilized paid personal caregivers (such as privately hired aides) and 0 otherwise. In Model 3, the dependent variable was total care expenditure, which measured the total amount of care-related spending in units of 10,000 KRW. Respondents who reported no care service utilization were coded as 0 KRW.

### 2.3. Independent Variables

The key independent variable in this study was the utilization of formal care services provided through the long-term care insurance (LTCI) system. This variable was coded as 1 for individuals who used LTCI-approved services, including home-based or institutional care, and 0 for those who did not utilize such services. In addition to formal care utilization, a range of other independent variables was included and categorized into two domains: public social security and private safety nets.

Within the domain of public social security, two separate binary variables were used to capture whether respondents received the basic pension and the national pension, respectively (each coded as 1 = recipient, 0 = non-recipient). For private safety nets, private transfer income was treated as a continuous variable indicating the amount of financial support received from family or other informal sources, and private health insurance was included as a binary variable (1 = enrolled, 0 = not enrolled).

### 2.4. Health-Related Factors

Health status was measured using four indicators. Subjective health was assessed on a 5-point Likert scale ranging from 1 (very poor) to 5 (very good). Disability status was coded as 1 for individuals who had received an official disability diagnosis and 0 otherwise. Activity limitation was included as a continuous variable, with higher scores indicating greater restriction in daily functioning due to health conditions. Finally, dementia diagnosis was captured as a binary variable, with 1 indicating a diagnosis of dementia or mild cognitive impairment and 0 indicating no such diagnosis.

### 2.5. Demographic Variables

Control variables included key demographic and socioeconomic characteristics. Gender was coded as 1 for male and 0 for female. Age was treated as a continuous variable ranging from 65 to 103 years. Marital status was included as a binary variable indicating the presence of a spouse (1 = has spouse, 0 = no spouse). Residential areas were categorized into three groups: rural areas (1), small and medium-sized cities (2), and metropolitan areas (3). Household assets were included as a continuous variable reflecting the respondent’s total household wealth

### 2.6. Statistical Analysis

To examine differences in caregiving type and financial burden among older adults, this study employed multiple analytical techniques. First, frequency analysis and descriptive statistics were conducted to summarize the characteristics of the study population and key variables. Second, group difference analyses, including *t*-tests, were performed to compare caregiving outcomes across groups. Additionally, correlation analysis was conducted prior to the regression analysis to check for multicollinearity, and no variables were found to have a correlation coefficient exceeding 0.8, indicating no serious multicollinearity concerns.

To identify factors associated with caregiving type and economic burden, two types of regression analyses were applied. Logistic regression analysis was used to estimate the likelihood of receiving family-provided care or private non-family care (binary outcomes), while linear regression analysis was used to assess predictors of total care expenditures (a continuous outcome variable). All models controlled for demographic, health-related covariates to ensure robust estimation.

## 3. Results

### 3.1. Demographic Factors

Table 1 presents the descriptive statistics of the study population (*n* = 3103), including demographic and health-related characteristics. The average age of respondents was 76.6 years. Female participants accounted for 60.8% of the sample, while males comprised 39.2%. Approximately two-thirds of respondents (68.6%) were living with a spouse.

In terms of residential areas, 30.9% resided in rural areas, 31.1% in small and medium-sized cities, and 38% in metropolitan regions. The average household assets were 39,545.6 thousand KRW with a large standard deviation, indicating considerable variability in wealth levels. The mean score for subjective health was 2.9 on a 5-point Likert scale, while the average level of activity limitation was 3.4. In terms of disability status, 0.7% of respondents had a disability. Limited Activity was measured as a continuous variable reflecting the degree of functional limitation in daily activities, with higher scores indicating greater restriction. The distribution across categories showed that 9.8% reported no limitation, 49.9% mild limitation, 32.8% moderate limitation, and 7.5% severe limitation. Additionally, 3% of the respondents were diagnosed with dementia or mild cognitive impairment, whereas the majority (97%) did not report cognitive issues.

A total of 14.5% of respondents received some form of care, while 86.8% did not receive any care services. Among those who received care, 8.5% relied solely on formal care (LTCI), and 86.1% received only family-provided care. Notably, only 2.7% of respondents received private non-family care.

### 3.2. t-Test

Table 2 presents the mean differences in caregiving types and total care expenses between older adults who utilized formal long-term care insurance (LTCI) services and those who did not. On average, individuals using LTCI services reported significantly higher levels of both family-provided care (M = 0.60, SD = 0.49) and private non-family care (M = 0.08, SD = 0.27) compared to those without LTCI utilization (family-provided care: M = 0.12, SD = 0.32; private non-family care: M = 0.00, SD = 0.07). Total care expenses were also substantially higher among LTCI users (M = 8.26, SD = 16.03) than non-users (M = 0.44, SD = 6.48). Independent sample *t*-tests confirmed that all differences were statistically significant at *p* < 0.001.

### 3.3. Crowding-In Effects

#### 3.3.1. Crowding-In Effects on Determinants of Family Caregiving

Our analysis, as presented in Table 3, reveals significant findings regarding the impact of long-term care insurance (LTCI) and related factors on family-provided care. Specifically, the study identifies a notable crowding-in effect, whereby increased utilization of formal LTCI services is positively associated with family-provided care. This suggests that public long-term care services tend to complement, rather than replace, family caregiving efforts.

Several public and private security factors were found to influence family-provided care. First, older adults receiving the basic pension were more likely to receive care from family members. Second, those who did not receive private transfer income were more likely to receive family care. Third, enrollment in private health insurance was associated with increased family caregiving.

Health status plays a crucial role in determining the level of family caregiving. First, lower subjective health status was associated with increased family caregiving. Second, greater limitations due to health issues correlated with more family-provided care. Lastly, diagnoses of dementia or mild cognitive impairment were linked to higher levels of family caregiving.

Age emerged as a significant demographic factor, with older individuals more likely to receive care from family members.

The observed crowding-in effect of LTCI on family care challenges the notion that public services replace family efforts. Instead, it suggests a complementary relationship that could be leveraged to enhance the overall quality and coverage of care.

#### 3.3.2. Crowding-In Effects on Determinants of Paid Caregiving

The study reveals a significant crowding-in effect. In particular, results from Model 2, as shown in Table 4, which examines caregiving provided by sources other than family or public services, show no statistically significant impact of formal care (LTCI). This suggests that the introduction of public long-term care services has neither a crowding-in nor a crowding-out effect on private non-family care provision. This finding implies that the private care sector operates independently of public care services, at least in the current Korean context.

Interestingly, neither the National Pension nor the Basic Pension showed a significant association with the use of private non-family care services. This suggests that these forms of old-age income security do not substantially influence the decision to use private care services.

Health status emerged as a significant predictor of private non-family care utilization. Individuals with greater health-related limitations were more likely to receive care from private, non-family caregivers. This indicates that as the level of care needs increases, there is a higher tendency to seek assistance from professional caregivers outside the family and public care systems.

Also, unmarried older adults were more likely to receive care from private, non-family caregivers. This finding highlights the importance of marital status in determining care arrangements, possibly due to the absence of a spouse as a potential caregiver.

The lack of significant crowding effects from public long-term care services on private care suggests a potential for complementary roles between public and private care sectors. Additionally, the findings underscore the importance of considering health status and marital status when designing and implementing long-term care policies.

#### 3.3.3. Crowding-In Effects on Determinants of Caregiving Expenditures

Table 5 presents the results of a multivariable linear regression analysis examining the determinants of total care expenditures. Utilization of formal care service through the long-term care insurance (LTCI) system was significantly associated with higher care expenditures, indicating that individuals who accessed LTCI-covered services incurred greater expenditures compared to non-users. This suggests that formal care does not replace private spending but instead complements it, reflecting a crowding-in effect in financial terms.

Private non-family care was found to have the largest impact on caregiving expenditures, highlighting the substantial financial burden associated with hiring personal caregivers. In contrast, family-provided care was not significantly associated with expenditure levels, suggesting that unpaid family care does not lead to additional measurable costs in the household budget.

Among the health-related variables, having a disability and dementia were both significantly associated with increased care expenditures, reflecting greater service needs and care intensity. Other variables, such as subjective health, activity limitations, demographic characteristics, and income security factors, were not significantly associated with care expenditures.

These findings reinforce the complementary, rather than substitutive, role of formal care services in Korea’s mixed care system and emphasize the disproportionate financial burden associated with paid caregiving and complex health conditions.

Figure 1 presents the estimated coefficients and 95% confidence intervals for the effects of key predictors on three outcome variables: family-provided care (M1, blue), private non-family care (M2, red), and total care expenditures (M3, green). The variables include long-term care insurance (LTC), basic pension, private transfer income, and private health insurance.

## 4. Discussion

This study provides comprehensive insights into the dynamics of long-term care for older adults in Korea, focusing on the interactions between public long-term care services, family-provided care, and private non-family care.

First, the study demonstrates a significant crowding-in effect of LTCI on family-provided care. Contrary to opinions that public services might substitute familial caregiving efforts, the results show that public services can complement family caregiving, supporting care arrangements rather than replacing them. Receipt of the basic pension, private transfer income, and enrollment in private health insurance were associated with greater reliance on family care. These findings suggest that financial and social resources tend to reinforce family caregiving rather than facilitate access to formal care services. This is consistent with findings from a similar study [8], which also reported a crowding-in effect on family care in China.

Health status emerged as another major determinant. Lower self-reported health status, greater health-related limitations, and diagnoses of dementia or mild cognitive impairment were all linked to higher levels of family-provided care. Furthermore, older age significantly increased the likelihood of receiving family care, underscoring the importance of addressing the evolving care needs of the oldest segments of the population.

In contrast, the analysis of private non-family care presented a different dynamic. The study found no significant relationship between LTCI utilization and the use of privately hired caregiving services, suggesting that the private care sector operates independently from public services. Similarly, the receipt of pensions, such as the national pension and the basic pension, showed no significant influence on private non-family caregiving decisions. Instead, health-related limitations and marital status were the primary drivers of private care utilization. Older adults with greater care needs and unmarried individuals were more likely to seek support from private, non-family caregivers, highlighting specific vulnerabilities within the aging population that require targeted policy attention.

The findings related to caregiving expenditures revealed that LTCI utilization was significantly associated with higher care costs. Rather than reducing financial burden, the use of public long-term care services was linked to increased expenditures. This suggests that LTCI may function as a complementary layer of care, adding to the overall intensity and complexity of care provision rather than substituting existing sources. These results reinforce the differentiated roles of public, family, and private caregiving, where formal services supplement, rather than replace, other forms of care—including financial inputs.

Taken together, these results emphasize the complexity of long-term care arrangements in Korea. The observed crowding-in effect on family caregiving demonstrates that public LTCI can reinforce familial care structures, while the independence of private caregiving suggests that the private care sector operates largely on a separate track, influenced by individual financial capacity and care preferences. Importantly, factors such as financial resources, health limitations, and demographic characteristics (e.g., marital status) significantly shape caregiving patterns and must be considered in policy design.

To ensure a resilient and sustainable long-term care system, Korea should pursue integrated strategies that balance the roles of public, private, and family caregiving sectors. Strengthening public support for lower- and middle-income groups—who may face financial barriers to accessing quality care—is critical to reducing reliance on family care driven by necessity rather than choice. This could include measures such as lowering the copayment rate for long-term care services, expanding the range of publicly covered services, and implementing targeted subsidies for vulnerable populations. At the same time, the public sector should play a stronger role in regulating and overseeing private care markets to ensure safety, quality, and fairness for those who use private services. Introducing a quality certification system for private care providers and establishing pricing transparency standards may help promote fairness and protect users.

The complementary relationship observed between public LTCI and family care likely reflects the limited scope of public care provision, which may not be sufficient to replace family efforts. However, this relationship should not be framed as a simple dichotomy of substitution versus complementarity. Rather, an appropriate mixture of public and private care is necessary, with public systems ensuring a basic level of access for those with fewer resources and private care serving as an additional option for those with greater means. By doing so, Korea can move toward a more comprehensive and equitable long-term care system capable of meeting the complex challenges of an aging society.

Moreover, as Korea continues to reform and expand its LTCI system, understanding these complex interactions will be critical for ensuring long-term sustainability. The Korean experience offers valuable insights that may inform similar efforts in other countries with aging populations, contributing to the broader global discourse on long-term care policy.

This study has several limitations. First, the analysis conducted a single wave of data, which limits the ability to establish causal relationships between LTCI utilization and caregiving patterns. While the study identifies associations, it cannot determine the directionality of these relationships, raising the possibility of reverse causality. Second, there may be potential selection bias, as individuals who utilize LTCI services may differ systematically from those who do not in terms of unobserved factors.

## 5. Conclusions

This study highlights the evolving dynamics of long-term care provision in Korea, demonstrating a crowding-in effect of LTCI on family caregiving and a largely independent role of private care services. These findings suggest that public services can reinforce, rather than displace, informal care networks. To build a more equitable and sustainable care system, policy efforts must focus on expanding public coverage, supporting family caregivers, and integrating private care in ways that complement public provision. Korea’s experience offers valuable lessons for other aging societies seeking to balance public, private, and familial care resources.

## Figures and Tables

**Figure 1 healthcare-13-01357-f001:**
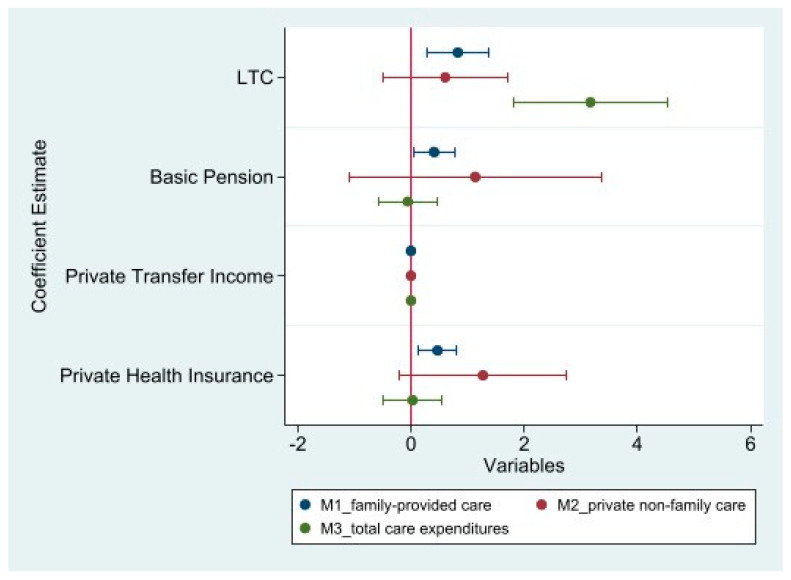
Regression coefficients and confidence intervals.

**Table 1 healthcare-13-01357-t001:** Demographic and health-related factors.

		Freq/Mean	Percent/Std. dev.
Gender	Female	1887	60.8
	Male	1216	39.2
Age	(Mean)	76.6	7.5
Spouse	Non-Spouse	973	31.4
	Spouse	2130	68.6
Residential Area	Rural	959	30.9
	Small and Medium	964	31.1
	Metropolitan	1180	38.0
Household Assets	(Mean)	39,545.6	50,224.3
Subjective Health	(Mean)	2.85	0.8
Disability	Without disability	3080	99.3
	With disability	23	0.7
Limited Activity	(Mean)	3103	3.4
	No or minimal limitation	304	9.8
	Mild limitation	1547	49.9
	Moderate limitation	1019	32.8
	Severe limitation	233	7.5
Dementia	Non-Dementia	3011	97.0
	Dementia	92	3.0
Care Receipt	Non-care	2652	86.8
	Any care	451	14.5
Type of Care	Only Formal	34	8.5
	Only Family-provided care	346	86.1
	Private non-family care	11	2.7

**Table 2 healthcare-13-01357-t002:** Caregiving types and expenditures by LTCI utilization (*n* = 3103).

	Non-Formal Caregiving (LTC)		Formal Caregiving (LTC)		t	*p*-Value
	Mean	Std. Dev.	Mean	Std. Dev.		
Family-provided care	0.12	0.32	0.60	0.49	−13.85	*p* < 0.001
Private non-family care	0.00	0.07	0.08	0.27	−8.20	*p* < 0.001
Total care expenses	0.44	6.48	8.26	16.03	−10.52	*p* < 0.001

**Table 3 healthcare-13-01357-t003:** Results of multivariable logistic regression on determinants of family-provided care (*n* = 3103).

	Coefficient	Standard Error	[95% Conf. Interval]		*p*-Value
Main variable					
Formal care (LTC)	0.83	0.28	0.28	1.37	0.003
National Pension	0.18	0.14	−0.09	0.44	0.193
Basic Pension	0.40	0.18	0.04	0.76	0.029
Private Transfer Income	0.00	0.00	0.00	0.00	0.018
Private Health Insurance	0.47	0.17	0.13	0.81	0.006
Health-related factors					−0.43
Health	−0.63	0.10	−0.82	−0.43	*p* < 0.001
Disability	1.06	0.55	−0.01	2.14	0.052
Limited Activity	0.66	0.10	0.46	0.85	*p* < 0.001
Dementia	1.55	0.28	1.00	2.11	*p* < 0.001
Demographics					0.48
Gender (ref = Female)	0.18	0.15	−0.12	0.48	0.25
Age	0.10	0.01	0.08	0.13	*p* < 0.001
Spouse	0.30	0.16	−0.01	0.62	0.061
Residential Area (ref = rural)					
Small and Medium	−0.05	0.16	−0.37	0.26	0.738
Metropolitan	0.15	0.15	−0.15	0.44	0.333
Household Assets	0.00	0.00	0.00	0.00	0.597
_cons	−11.70	1.10	−13.86	−9.55	*p* < 0.001

Log likelihood = −872.6643, LR chi2(15) = 654.91, Prob > chi2 = 0.0000, Pseudo R^2^ = 0.2729.

**Table 4 healthcare-13-01357-t004:** Results of multivariable logistic regression on determinants of private non-family care (*n* = 3103).

	Coefficient	Standard Error	[95% Conf. Interval]		*p*-Value
Main variable					
Formal care (LTC)	0.61	0.56	−0.48	1.70	0.276
National Pension	−0.39	0.55	−1.47	0.69	0.481
Basic Pension	1.13	1.13	−1.09	3.35	0.319
Private Transfer Income	1.27	0.75	−0.20	2.75	0.091
Private Health Insurance	0.00	0.00	0.00	0.00	0.012
Health-related factors					
Health	−0.38	0.40	−1.18	0.41	0.343
Disability	0.01	1.16	−2.27	2.29	0.992
Limited Activity	1.74	0.54	0.68	2.79	0.001
Dementia	1.04	0.57	−0.08	2.16	0.069
Demographics					
Gender (ref = Female)	0.51	0.62	−0.71	1.72	0.414
Age	0.03	0.04	−0.05	0.11	0.453
Spouse	−1.36	0.67	−2.68	−0.04	0.044
Residential Area(ref = rural)					
Small and Medium	−0.76	0.64	−2.01	0.50	0.237
Metropolitan	−0.23	0.54	−1.29	0.84	0.678
Household Assets	0.00	0.00	0.00	0.00	0.738
_cons	−14.56	4.50	−23.38	−5.74	0.001

Log likelihood = −84.4, LR chi2(15) = 92.80, Prob > chi2 = 0.0000, Pseudo R^2^ = 0.3547.

**Table 5 healthcare-13-01357-t005:** Results of multivariable linear regression on determinants of total care expenditures (*n* = 3103).

	Coefficient	Standard Error	[95% Conf. Interval]		*p*-Value
Main variable					
Formal care (LTC)	3.17	0.69	1.82	4.53	*p* < 0.001
Family-provided care	0.15	0.36	−0.56	0.86	0.686
Private non-family care	40.71	1.30	38.16	43.27	*p* < 0.001
National Pension	0.21	0.23	−0.24	0.66	0.355
Basic Pension	−0.06	0.27	−0.58	0.46	0.825
Private Transfer Income	0.03	0.27	−0.49	0.55	0.908
Private Health Insurance	0.00	0.00	0.00	0.00	*p* < 0.001
Health-related factors					
Health	−0.31	0.16	−0.63	0.01	0.056
Disability	3.57	1.26	1.09	6.04	0.005
Limited Activity	0.07	0.17	−0.26	0.40	0.681
Dementia	1.86	0.69	0.50	3.21	0.007
Demographics					
Gender (ref = Female)	0.06	0.26	−0.45	0.56	0.829
Age	0.03	0.02	−0.01	0.07	0.113
Spouse	0.27	0.28	−0.28	0.81	0.334
Residential Area(ref = rural)					
Small and Medium	0.22	0.27	−0.32	0.76	0.422
Metropolitan	−0.23	0.26	−0.74	0.29	0.386
Household Assets	0.00	0.00	0.00	0.00	0.850
_cons	−2.24	1.79	−5.75	1.28	0.213

F (17, 3085) = 77.21, Prob > F= 0.0000, R^2^ = 0.2985, Adj R^2^ = 0.2946.

## Data Availability

The data used in this study were obtained from the Korean Longitudinal Study of Ageing (KLoSA), managed by the Korean Employment Information Service (https://survey.keis.or.kr, accessed on 29 April 2025). Data access for analysis purposes was first initiated on 8 July 2024, and the latest access date for verification was 29 April 2025.

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
