# Peer review of "The Impact of Long-Term Care Insurance for Older Adults: Evidence of Crowding-In Effects"

_healthcare, 2025, doi:10.3390/healthcare13121357_

Round 1
Reviewer 1 Report
Comments and Suggestions for Authors
Thank you for providing this insight into your work: I believe this deserves to be published, but you need to improve it, in my view. Some minor points, and some suggestions for improvement.
10 "private systems" too vague
12, 176-7 etc. "rise": suggests dynamics and that you follow these individuals, but I understand that this is a cross-sectional snapshot, where you use that survey? You will know if they use more or less, but not if there is a "rise"...? My belief from what I have seen elsewhere is that this is a gradual process over time, supplementing family care w public services etc. where some people willingly do it, others refuse to use public services and/or help from their family etc.
16 have > has
90 - 131 - 135-6 and other places: "caregiving landscape" etc. But I understand this is a survey of (potential) USERS of care (care recipients) rather than givers...
And in that context: just a few per cent are disabled and/or demented - is that likely? or has the survey not targeted disabled, sick etc. people?
And reporting data: you provide ridiculously exaggerated exactness with (line 147) subjects being 76.58 years, 60.81 % being women etc. Suggest that you round of figures (to 77 or possibly 76.6 years etc.). If you have a census of all (South) Koreans you can be that exact...
The demographic data give the impression that all these 65-103 year olds are so well and healthy and need so little care: Then we wonder how you can analyse the various constellations of family-public-private care... I guess that several more than in your data have some disablility and need some help etc. but this should be reported with percentages etc. on the various indicators?
And one last point: the manuscript says that you accessed the data April 29: Very rapid analysis!
Good Luck with your work, I look forward to a revised version
193 "---services necessarily replace family efforts---" necessarily?
The manuscript also should report % who need care (if available) and the resp. percentages who receive
1 family care
2 private care (aides etc.)
3 formal care LTCI etc.
and the overlap between these three caregivers (% who receive care from both family and LTCI etc.)
Comments on the Quality of English Language
Needs some improvement also.
Author Response
Comments 1 : Thank you for providing this insight into your work: I believe this deserves to be published, but you need to improve it, in my view. Some minor points, and some suggestions for improvement.
Response 1 : Thank you for your thoughtful and constructive comments. I have carefully considered your suggestions, and I believe they have significantly improved the quality and clarity of the manuscript.
Comments 2 :10 "private systems" too vague
Response 2 : Thank you for your valuable feedback. To clarify, I have revised the sentence to explicitly specify the components of private systems as private transfer income and private health insurance. The revised sentence is below.
Additionally, it examines the influence of old-age income security and private systems, including private transfer income and private health insurance, on these effects.
Comments 3 : 12, 176-7 etc. "rise": suggests dynamics and that you follow these individuals, but I understand that this is a cross-sectional snapshot, where you use that survey? You will know if they use more or less, but not if there is a "rise"...? My belief from what I have seen elsewhere is that this is a gradual process over time, supplementing family care w public services etc. where some people willingly do it, others refuse to use public services and/or help from their family etc.
Response 3 : Thank you for your valuable feedback regarding the use of the term “rise” and the clarification that this is a cross-sectional study. To address this, I have revised the sentence to more accurately reflect the association, avoiding causal implications. The revised sentences are as follows:
The results reveal a crowding-in effect for family care, as greater utilization of public LTCI is positively associated with family-provided care.
Specifically, the study identifies a notable crowding-in effect, whereby increased utilization of formal LTCI services is positively associated with family-provided care.
Comments 4 :16 have > has
Response 4 : Thank you for pointing out the typographical error. I have corrected the sentence as follows:
It is noteworthy that the current public LTCI in Korea has low coverage, resulting in insufficient care provision.
Comments 5 : 90 - 131 - 135-6 and other places: "caregiving landscape" etc. But I understand this is a survey of (potential) USERS of care (care recipients) rather than givers...
Response 5 : Thank you for your insightful comment. I originally used the term “caregiving landscape” to refer to the overall care environment and systems, but I understand that this may cause confusion when referring specifically to care recipients rather than caregivers. To ensure greater clarity, I have revised the relevant sentences as follows:
Understanding the mechanisms of crowding-in and crowding-out is central to analyzing the evolving landscape of LTC systems.
With the introduction of public long-term care insurance (LTCI) in Korea in 2008, the care system for older adults has undergone a significant shift.
This investigation contributes to a deeper understanding of the crowding-in and crowding-out effects of formal care services within Korea’s long-term care system.
Comments 6 : And in that context: just a few per cent are disabled and/or demented - is that likely? or has the survey not targeted disabled, sick etc. people?
Response 6 : Thank you for your thoughtful question. The survey defines “disabled” individuals specifically as those who have been officially approved for a disability registration, meaning that individuals with functional limitations who do not meet this strict criterion are not classified as disabled in the data. Additionally, the data I used primarily targets community-dwelling older adults in Korea, resulting in a sample that predominantly includes relatively healthy individuals, rather than those with severe illnesses or disabilities.
Comments 7 : And reporting data: you provide ridiculously exaggerated exactness with (line 147) subjects being 76.58 years, 60.81 % being women etc. Suggest that you round of figures (to 77 or possibly 76.6 years etc.). If you have a census of all (South) Koreans you can be that exact...
Response 7 :Thank you for your valuable suggestion. I have adjusted the reporting of numerical data in Table 1. Demographic and Health-related Factors by rounding figures to one decimal place for clarity and readability.
Comments 8 : The demographic data give the impression that all these 65-103 year olds are so well and healthy and need so little care: Then we wonder how you can analyse the various constellations of family-public-private care... I guess that several more than in your data have some disablility and need some help etc. but this should be reported with percentages etc. on the various indicators?
Response 8 : It is true that the demographic data may give the impression that the study participants are generally well and require little care. However, when examining limited Activity, which reflects self-reported functional limitations in daily life, a different picture emerges. Specifically, while 9.8% reported no or minimal limitation, 49.9% reported mild limitation, 32.8% moderate limitation, and 7.5% severe limitation. This distribution indicates that a substantial proportion of older adults experience some degree of functional limitation that may necessitate care. Previously, limited Activity was presented as a continuous variable, but it is now reported as categorical frequencies to better reflect the distribution of care needs within the study population in Table 1.
Comments 9 : the manuscript says that you accessed the data April 29: Very rapid analysis!
Response 9 : I initially indicated the most recent date I accessed the data, which was April 29, 2025. However, for clarity, I confirm that the earliest data access for this research was on July 8, 2024. The April 29 date refers to the final access for verification prior to submission. To ensure clarity, I have presented both dates in the Data Availability Statement.
Data Availability Statement: The data used in this study were obtained from the Korean Longitudinal Study of Ageing (KLoSA), managed by the Korean Employment Information Service (https://survey.keis.or.kr). Data access for analysis purposes was first initiated on July 8, 2024, and the latest access date for verification was April 29, 2025.
Comments 10 : 193 "---services necessarily replace family efforts---" necessarily?
Response 10 : Thank you for your comment. I have removed the word "necessarily," as it was unnecessary in the context. The revised sentence is as follows:
The observed crowding-in effect of LTCI on family care challenges the notion that public services replace family efforts.
Comments 11 : The manuscript also should report % who need care (if available) and the resp. percentages who receive
1 family care
2 private care (aides etc.)
3 formal care LTCI etc.
and the overlap between these three caregivers (% who receive care from both family and LTCI etc.)
Response 11 : Thank you for your feedback. I have now included the relevant information in Table 1 and the text.
14.5% of respondents received some form of care, while 86.8% reported did not receive any care services. Among those who received care, 8.5% relied solely on formal care (LTCI), 86.1% received only family-provided care. Notably, only 2.7% of respondents received private non-family care.
Reviewer 2 Report
Comments and Suggestions for Authors
It is recommended that the introduction include more national data on the historical evolution of LTCI in Korea, such as the number of beneficiaries year by year, and clarify the conceptual differences between “crowding-in” and “crowding-out” with concrete examples.
In the methodology section, it would be advisable to detail the exclusion criteria, indicating how many cases were removed and why; in addition, clarify whether multicollinearity among independent variables was tested.
Regarding the results, it is necessary to include confidence intervals in the regression tables in order to assess the precision of the estimates. It is also recommended to add a bar chart to visualize key coefficients (LTC, basic pension, activity limitation).
As for the discussion and conclusions, it is essential to include a discussion on the limitations of the cross-sectional design, such as reverse causality and the potential for selection bias. Furthermore, the authors should include specific policy proposals to expand public coverage, for example, by suggesting a recommended copayment percentage. The inclusion of studies published after 2020 is also encouraged, in order to compare the results.
Author Response
Comments 1 : It is recommended that the introduction include more national data on the historical evolution of LTCI in Korea, such as the number of beneficiaries year by year, and clarify the conceptual differences between “crowding-in” and “crowding-out” with concrete examples.
Response 1 : Thank you for your valuable comments. In response, I have incorporated national data on the historical evolution of Korea’s Long-Term Care Insurance (LTCI) system, as well as clarified the conceptual differences between “crowding-in” and “crowding-out” with concrete examples in the revised manuscript.
In Korea, the public LTCI system was introduced in July 2008 to address the rapidly growing needs of its aging population. The number of individuals certified for LTCI benefits grew from 770,000 in 2019 to 860,000 in 2020, 950,000 in 2021, and exceeded 1,010,000 in 2022. In 2023, the number of recipients who actually used LTCI services reached 1,073,452, marking a 7.4% increase from the previous year [6].
"Crowding-out" refers to situations where the introduction or generosity of public LTCI leads to a reduction in informal caregiving efforts, typically as formal care substitutes for previously unpaid familial support [8, 10]. For example, families may reduce the intensity of caregiving, particularly for physically demanding tasks such as bathing, transferring, and toileting—after their family member begins using LTCI services. In contrast, "crowding-in" occurs when formal LTC programs enhance informal caregiving by alleviating caregiver burdens, expanding the pool of caregivers, or improving the sustainability of family support networks [7, 11]. For example, the availability of formal LTCI services may enable families to continue providing emotional support and household assistance, while relying on professional caregivers for more intensive tasks, thereby sustaining overall caregiving involvement.
Comments 2 : In the methodology section, it would be advisable to detail the exclusion criteria, indicating how many cases were removed and why; in addition, clarify whether multicollinearity among independent variables was tested.
Response 2 : Thank you for your helpful suggestions. In response, I have revised the methodology section to clarify the exclusion criteria. Additionally, I have clarified in the text that multicollinearity among independent variables was tested using correlation analysis, and no variables were found to have a correlation coefficient exceeding 0.8, indicating no serious multicollinearity concerns.
Of the total 6,057 respondents, individuals under the age of 65 (n=1,566) and those with missing values in key variables for this analysis (n=1,388) were excluded, resulting in a final sample of 3,103 individuals aged 65 and older.
Additionally, correlation analysis was conducted prior to the regression analysis to check for multicollinearity, and no variables were found to have a correlation coefficient exceeding 0.8, indicating no serious multicollinearity concerns.
Comments 3 : Regarding the results, it is necessary to include confidence intervals in the regression tables in order to assess the precision of the estimates. It is also recommended to add a bar chart to visualize key coefficients (LTC, basic pension, activity limitation).
Response 3 : Thank you for your feedback. In response, I have added confidence intervals to the regression tables to improve the precision and interpretability of the estimates. Additionally, I have included a chart visualizing key coefficients.
Comments 4 : As for the discussion and conclusions, it is essential to include a discussion on the limitations of the cross-sectional design, such as reverse causality and the potential for selection bias.
Response 4 : Thank you for your feedback. I have included a discussion of the study’s limitations.
This study has several limitations. First, the analysis conducted a single wave of data, which limits the ability to establish causal relationships between LTCI utilization and caregiving patterns. While the study identifies associations, it cannot determine the directionality of these relationships, raising the possibility of reverse causality. Second, there may be potential selection bias, as individuals who utilize LTCI services may differ systematically from those who do not, in terms of unobserved factors.
Comments 5 : Furthermore, the authors should include specific policy proposals to expand public coverage, for example, by suggesting a recommended copayment percentage.
Response 5 : Thank you for the suggestion. I have added more specific policy recommendations. However, determining an optimal copayment rate requires further research tailored to Korea’s specific context. Therefore, I have not added a specific percentage in this paper.
To ensure a resilient and sustainable long-term care system, Korea should pursue integrated strategies that balance the roles of public, private, and family caregiving sectors. Strengthening public support for lower- and middle-income groups—who may face financial barriers to accessing quality care—is critical to reducing reliance on family care driven by necessity rather than choice. This could include measures such as lowering the copayment rate for long-term care services, expanding the range of publicly covered services, and implementing targeted subsidies for vulnerable populations. At the same time, the public sector should play a stronger role in regulating and overseeing private care markets to ensure safety, quality, and fairness for those who use private services. Introducing a quality certification system for private care providers and establishing pricing transparency standards may help promote fairness and protect users.
Comments 6 : The inclusion of studies published after 2020 is also encouraged, in order to compare the results.
Response 6 : Thank you for the suggestion. While there are relatively few recent articles focusing specifically on crowding-in and crowding-out effects, I have included relevant studies published after 2020, such as a 2023 study, to provide a more up-to-date comparison. I have also incorporated these studies into the results section for comparison, such as the following:
This is consistent with findings from a similar study [8], which also reported a crowding-in effect on family care in China.
Round 2
Reviewer 1 Report
Comments and Suggestions for Authors
Hi
line 168 change >> reported no limitation
line 172 delete "reported"
Author Response
Comments:
line 168 change >> reported no limitation
line 172 delete "reported"
Response: Thank you for your feedback. I have revised the manuscript as per your suggestions
The distribution across categories showed that 9.8% reported no limitation, 49.9% mild limitation, 32.8% moderate limitation, and 7.5% severe limitation.
14.5% of respondents received some form of care, while 86.8% did not receive any care services.